# Comparative Analysis of Single-Molecule Dynamics of TRPV1 and TRPV4 Channels in Living Cells

**DOI:** 10.3390/ijms22168473

**Published:** 2021-08-06

**Authors:** Yutaro Kuwashima, Masataka Yanagawa, Mitsuhiro Abe, Michio Hiroshima, Masahiro Ueda, Makoto Arita, Yasushi Sako

**Affiliations:** 1Cellular Informatics Laboratory, RIKEN Cluster for Pioneering Research, 2-1 Hirosawa, Wako 351-0198, Saitama, Japan; yutaro-765@keio.jp (Y.K.); abemitsu@riken.jp (M.A.); m_hiroshima@riken.jp (M.H.); 2Division of Physiological Chemistry and Metabolism, Keio University Faculty of Pharmacy, Tokyo 105-0011, Japan; arita-mk@pha.keio.ac.jp; 3Japan Science and Technology Agency (JST), PRESTO, 4-1-8, Honcho, Kawaguchi 332-0012, Saitama, Japan; 4Laboratory for Cell Signaling Dynamics, RIKEN Center for Biosystems Dynamics Research (BDR), 6-2-3, Furuedai, Suita 565-0874, Osaka, Japan; masahiroueda@fbs.osaka-u.ac.jp; 5Laboratory of Single Molecule Biology, Graduate School of Frontier Biosciences, Osaka University, 1-3 Yamadaoka, Suita 565-0871, Osaka, Japan; 6Laboratory for Metabolomics, RIKEN Center for Integrative Medical Sciences (IMS), Yokohama 230-0045, Kanagawa, Japan; 7Cellular and Molecular Epigenetics Laboratory, Graduate School of Medical Life Science, Yokohama City University, Yokohama 230-0045, Kanagawa, Japan

**Keywords:** TRPV channel, single-molecule imaging, diffusion, endocytosis, receptor oligomerization

## Abstract

TRPV1 and TRPV4, members of the transient receptor potential vanilloid family, are multimodal ion channels activated by various stimuli, including temperature and chemicals. It has been demonstrated that TRPV channels function as tetramers; however, the dynamics of the diffusion, oligomerization, and endocytosis of these channels in living cells are unclear. Here we undertook single-molecule time-lapse imaging of TRPV1 and TRPV4 in HEK 293 cells. Differences were observed between TRPV1 and TRPV4 before and after agonist stimulation. In the resting state, TRPV4 was more likely to form higher-order oligomers within immobile membrane domains than TRPV1. TRPV1 became immobile after capsaicin stimulation, followed by its gradual endocytosis. In contrast, TRPV4 was rapidly internalized upon stimulation with GSK1016790A. The selective loss of immobile higher-order oligomers from the cell surface through endocytosis increased the proportion of the fast-diffusing state for both subtypes. With the increase in the fast state, the association rate constants of TRPV1 and TRPV4 increased, regenerating the higher-order oligomers. Our results provide a possible mechanism for the different rates of endocytosis of TRPV1 and TRPV4 based on the spatial organization of the higher-order structures of the two TRPV channels.

## 1. Introduction

Transient receptor potential vanilloid 1 (TRPV1) and TRPV4 are members of the TRPV subfamily of polymodal nonselective cation channels. TRPV1 is expressed in the peripheral and central nervous systems and is involved in nociception [1]. It is activated by natural vanilloid compounds (capsaicin and resiniferatoxin (RTX)), noxious thermal stimuli (>43 °C), and protons. TRPV4 is broadly expressed in the human body, including in the central nervous and cardiovascular systems, and is involved in the regulation of various sensations and vascular tone [2]. TRPV4 is activated by osmotic stress, non-noxious thermal stimuli (30–34 °C), and several chemical compounds, including 4α-phorbol 12,13-didecanoate (4α-PDD), and GSK1016790A.

Previous structural analyses revealed that TRP channels, including TRPV1 and TRPV4, form tetrameric structures [3,4,5]. Each subunit consists of a six-transmembrane region and intracellular N- and C-termini. The organization of the tetrameric form of the channels in the plasma membrane is critical for their function. However, the dynamics of the TRPV channels in living cells, including their oligomerization, diffusion, and endocytosis, are yet to be clarified. A previous study using a number and brightness (N&B) analysis, which measures the apparent brightness and number of molecules, suggested that TRPV1 exists as dimers and that RTX stimulation promotes tetramer formation [6]. Total internal reflection fluorescence microscopy (TIRFM) showed that the mobility of TRPV1 in the plasma membrane decreases immediately after stimulation with capsaicin in the presence of extracellular calcium ions [7]. However, it is currently unclear whether these changes in dynamics upon activation are shared by other TRPV channels. There is also little knowledge of the changes in their dynamics over time after stimulation.

Here we investigated the time-dependent changes in the single-molecule dynamics of TRPV1 and TRPV4 before and after stimulation with an agonist. The single-molecule tracking analysis and a variational Bayesian hidden Markov model (VB-HMM) clustering analysis of their trajectories [8,9,10] revealed different features of the single-molecule behaviors of the TRPV1 and TRPV4 channels. Before agonist stimulation, the TRPV4 channels were more likely to form tetramers and higher-order oligomers than were the TRPV1 channels, which were confined to ~150 nm membrane domains. After agonist stimulation, TRPV4 showed rapid endocytosis (within 5 min), whereas it took >10 min for the internalization of TRPV1 to commence. During endocytosis, the immobile higher-order oligomers were selectively internalized into cells, and the proportion of fast-diffusing channels increased for both subtypes. The increasing proportion of the fast mobile state increased the probability of encounters between the subunits of the TRPV channels, enhancing the regeneration of tetramers.

## 2. Results

### 2.1. Time-Lapse Single-Molecule Imaging of HaloTag-Fused TRPV1 and TRPV4

To ensure quantitative single-molecule measurements, we first evaluated the endogenous TRPV1 and TRPV4 expression in HEK293 cells with a FLIPR calcium mobilization assay. GSK1016790A (TRPV4-specific agonist) stimulation significantly induced Ca^2+^ influx into the parental HEK293 cells (EC_50_ = 34 nM), in contrast to capsaicin (TRPV1-specific agonist) stimulation (Figure 1A,B), which was consistent with the previous report [11]. Therefore, we constructed a TRPV4-deficient HEK293 cell line using the CRISPR/Cas9 system (Figure 1C) and confirmed that no significant Ca^2+^ influx occurred in the TRPV4-knockout (KO) cell line upon GSK1016790A stimulation (Figure 1A). All of the subsequent experiments were performed with the TRPV4-KO cell line. We also confirmed that fusion of the HaloTag to the N-termini of TRPV1 and TRPV4 did not impair their channel activity (Figure 1D–I). The EC_50_ values were 11 nM of capsaicin and 26 nM of GSK1016790A on the HaloTag-fused TRPV1- and TRPV4-expressing HEK293 cells, respectively, which were approximately similar to previously reported values (28.5 nM of capsaicin in HEK293 cells [12], and 24 nM of GSK1016790A in human dermal fibroblasts [13]). HaloTag fusion allowed the specific labeling of TRPV channels with a bright, highly photostable fluorescent dye suitable for long-term single-molecule imaging.

We expressed HaloTag-fused TRPV1 or TRPV4 in the HEK293 cell line and stained them simultaneously with one of two fluorescent dyes (30 nM Janelia Fluor 549 HaloTag Ligand (JF549) and 30 nM HaloTag STELLA Fluor 650 ligand (SF650)) to measure the association and dissociation dynamics between the subunits of the TRPV channels. Based on a saturation binding assay, ~50% of the HaloTag-fused TRPV channels were labeled with 30 nM HaloTag ligand (Appendix A). Therefore, in total, >80% of the channels were labeled with one of the dyes under the present staining conditions.

We then examined the TRPV1 or TRPV4 channels with single-molecule time-lapse imaging at 5 min intervals for 20 min (Figure 2A,B, Appendix A). At each time point, we took 100-frame videos at a 33 frame per second. Upon agonist stimulation, a reduction in particle density was observed for both TRPV1 and TRPV4, reflecting their endocytosis, although the kinetics of the two molecules clearly differed. The reduction in the particle density of TRPV1 was first observed 10 min after capsaicin stimulation (Figure 2C and Appendix A). In contrast, the particle density of TRPV4 began to decrease within 5 min of GSK1016790A stimulation (Figure 2D and Appendix A). Because no significant changes were observed upon stimulation with the vehicle (Appendix A–D), the reduction in particle density was not attributable to the photobleaching or mechanical activation of the TRPV channels caused by the added solution.

We performed a single-molecule tracking analysis and compared the time- and-ensemble-averaged mean squared displacements (MSDs) of the total trajectories of the TRPV channels (Figure 2E,F and Appendix A). Consistent with a previous report [7], the MSD of TRPV1 decreased within 5 min of capsaicin stimulation. However, 10 min after stimulation, the MSD returned to the basal level and increased further over time (Figure 2E and Appendix A). In contrast, the MSD of TRPV4 increased about two-fold within 5 min of GSK1016790A stimulation, then remained at the higher level (Figure 2F and Appendix A). Vehicle stimulation did not change the MSD of either TRPV channel (Appendix A). The kinetics of endocytosis and the diffusion changes were partly correlated for both the TRPV1 and TRPV4 channels (Figure 2G,H and Appendix A).

### 2.2. Clustering Analysis of the Diffusion States of TRPV1 and TRPV4 Channels Based on a Three-State Hidden Markov Model

We clustered the diffusion states of the trajectories based on a variational Bayesian hidden Markov model (VB-HMM) [8]. Among the one- to five-state models, the three- and four-state models showed the highest lower bound values for both the TRPV1 and TRPV4 trajectories (Appendix A–D). To compare the diffusion dynamics of the TRPV channels on the same platform, we chose the three-state model (immobile, medium, and fast states in ascending order of diffusion coefficients), in which ~80% of the cells showed the highest lower bound. The trajectories and step-size histograms of the TRPV channels in each diffusion state are exemplified in Figure 3A–D. The diffusion coefficients were estimated to be in the ranges of 0.0077–0.0087 μm^2^/s (immobile), 0.039–0.054 μm^2^/s (medium), and 0.19–0.30 μm^2^/s (fast). The MSD–Δt plots of the fast and medium states for both channels were almost linear on average, indicating their simple diffusion (Figure 3E,F, left, Appendix A, left). In contrast, the MSD–Δt plots of the immobile state for both channels showed a convex upward shape on average (Figure 3E,F, right, Appendix A, right), suggesting that they contain molecules that are undergoing confined diffusion in membrane domains of ~150 nm.

To evaluate the relationship between the diffusion and oligomeric states of the TRPV channels, we next analyzed the intensity histograms in each diffusion state. The mean fluorescence intensity of the single molecules was estimated to be 890 arbitrary units (a.u.) for JF549 and 720 a.u. for SF650 from the fluorescence intensity histogram for HaloTag-fused CD86, a monomeric membrane protein (Figure 3G and Appendix A) [14]. The peak fluorescence intensities of the fast, medium, and immobile states of the TRPV channels were approximately two, three, and four times the single-molecule intensity, respectively (Figure 3H,I and Appendix A). Therefore, the particles were composed mainly of apparent dimers to tetramers. All the intensity histograms were broad and shifted to the right, suggesting that the TRPV channels form tetramers or higher-order oligomers to some extent even before agonist stimulation. The slower diffusion state contained a larger fraction of higher-order oligomers. TRPV4 showed a broader intensity histogram than TRPV1, suggesting a greater tendency to form higher-order structures.

### 2.3. Agonist-Induced Changes in Membrane Dynamics of the TRPV1 and TRPV4 Channels

To test the relationship between the endocytosis and diffusion states of TRPV1 and TRPV4, we analyzed the density changes of the particles in each diffusion state after agonist stimulation (Figure 4A,B). Within 5 min of capsaicin stimulation, the density of the immobile state of TRPV1 tended to increase, and the density of every state had decreased 10 min after stimulation (Figure 4A). When evaluated as the proportions of the diffusion states at each time point, the increase in the immobile state fraction was statistically significant and continued throughout the period 5–20 min after stimulation (Figure 4C). The reduction in the proportion of the medium state was significant 5 min later than was the change in the immobile state and was accompanied by an increase in the proportion of the fast state. Capsaicin stimulation did not change the average diffusion coefficient for each diffusion state (Appendix A). Therefore, the changes in the averaged MSDs in Figure 2E are attributable mainly to the changes in diffusion state ratios. We estimated the rate constants for the transitions between the three states from the transition matrix of HMM (Appendix A). Five minutes after stimulation, the transition rate from a faster to a slower state increased (Appendix A). The transitions from the fast state to the medium and immobile states then decreased 10 min after stimulation. In contrast, the transition from the medium to the immobile state continued to increase. To investigate the agonist-dependent changes in the oligomeric states, we analyzed the difference in the intensity histograms before and after stimulation (Figure 4G and Appendix A). The intensity histograms of the medium state of TRPV1 were particularly affected by the capsaicin treatment, with an increase in apparent dimers and a reduction in tetramers and higher-order oligomers after 10–20 min, rather than after 5 min. The immobile and fast states did not change much compared with the medium state. These results suggest that the TRPV1 channels accumulated in an immobile membrane domain, such as caveolae or clathrin-coated pits, within 5 min of activation, which was followed by endocytosis.

In contrast, TRPV4 showed a significant reduction in particle density in the medium and immobile states 5 min after GSK1016790A stimulation, whereas that of the fast state did not change during measurement (Figure 4B). The selective loss of the immobile and medium states from the cell surface caused an increase in the fast mobile state ratio (Figure 4D). GSK1016790A stimulation increased the average diffusion coefficients for the fast and medium states of TRPV4 (Appendix A). The transition rate constants from the fast state to the medium and immobile states decreased 5 min after GSK1016790A stimulation (Appendix A). These observations are similar to the changes in the transition rate constants of the TRPV1 channels between 5 and 10 min after capsaicin stimulation when endocytosis commenced (Appendix A). Similar to TRPV1, the intensity of the medium state of TRPV4 was altered by the agonist, although this change occurred within 5 min (Figure 4H and Appendix A). All the agonist-dependent changes in the TRPV channels described above were common to both fluorophores and were not observed after stimulation with vehicle (Appendix A). Notably, a larger fraction of the TRPV4 channels was immobile before stimulation than in the TRPV1 channels (Figure 4E,F). The pre-accumulation of the inactive TRPV4 channels in membrane domains may be responsible for the rapid endocytosis of TRPV4 upon its activation.

### 2.4. Colocalization Analysis of TRPV1 and TRPV4 upon Activation

To quantify the time-dependent changes in the association and dissociation dynamics between the subunits containing two or more molecules of the TRPV channels, we performed a colocalization analysis of the particles labeled with JF549 and SF650 (Figure 5A–D). Despite the change in particle density, the percentage of colocalized particles was ~20% of the total particles for both TRPV1 and TRPV4 (Figure 5E,F and Appendix A). The on-rate constants of the TRPV1 subunits did not change with the addition of vehicle, whereas in contrast, it increased gradually over time 10 min after capsaicin stimulation (Figure 5G). The on-rate constants of the TRPV4 particles increased significantly within 5 min of GSK1016790A stimulation, followed by a gradual increase over time (Figure 5H). In contrast, the on-time distributions of the colocalized subunits did not change after stimulation with agonist or vehicle (Figure 5I,J and Appendix A). These results suggest that the probability of encounters between subunits was increased by the increased proportion of the fast state.

## 3. Discussion

In the present study, we compared the time-dependent changes in the single-molecule dynamics of the TRPV1 and TRPV4 channels before and after stimulation with an agonist, including the particle density in the plasma membrane, the diffusion state ratio, the intensity distribution, and the association and dissociation rates between subunits.

Our results suggest the model shown in Figure 6 and Appendix A. Both TRPV1 and TRPV4 are in equilibrium among the three diffusion states (immobile, medium, and fast) in the plasma membrane. The faster diffusion state contains more dimers, and the slower diffusion states contain more tetramers and higher-order oligomers. Compared with TRPV1, more TRPV4 channels are confined in an immobile membrane domain in the inactive state. Because endocytosis occurs from membrane domains, such as caveolae and clathrin-coated pits, the pre-accumulation of TRP channels into immobile domains probably contributes to their rapid endocytosis. In fact, the endocytosis of TRPV4 occurs significantly faster than that of TRPV1 (Figure 2A–D). The entrapment of TRPV1 into the immobile domain occurs 5 min after its stimulation with capsaicin, followed by its endocytosis 10 min after capsaicin stimulation (Figure 4A). In contrast, TRPV4 was internalized within 5 min (Figure 4B). The increases in proportions of the fast state of both TRPV1 and TRPV4 were associated with the selective endocytosis of the immobile higher-order oligomers (Figure 4C,D). The growing proportion of the fast state increased the association rate constant between the dimeric channels (Figure 5G,H).

The 300 nM agonists used in the present study are the saturation concentration in the calcium mobilization assay (Figure 1F,I). In the previous studies, the EC_50_ values of GSK1016790A on human TRPV4 expressing HEK293 cells were reported to be 2.1 nM [15] and 5 nM [11] using a similar method. The apparent discrepancy in EC_50_ values of GSK1016790A between the present (26 nM) and the previous studies would be due to the expression level of TRPV4 channels. In general, overexpression of receptors increases the number of spare receptors, which may lead to a decrease in EC_50_ [16]. The dose-response curves of capsaicin and GSK1016790A on TRPV1- and TRPV4-transfected HEK293 cells were compared with those on mock-transfected HEK293 cells in the previous study [11]. Similar to Figure 1A, GSK1016790A induced the Ca^2+^ influx in the mock-transfected HEK293 cells [11]. The EC_50_ of TRPV4-transfected HEK293 cells was approximately 10-fold lower than that of mock-transfected HEK293 cells [11], suggesting that the overexpression of TRPV4 enhanced the apparent potency of GSK1016790A. Since the expression level of HaloTag-fused TRPV4 was suppressed for single-molecule imaging here, the EC_50_ of GSK1016790A on HaloTag-fused TRPV4-expressing HEK293 cells (26 nM) could become comparable to that on wild-type HEK293 cells (34 nM). These values were also in agreement with the previously reported EC_50_ of 24 nM in human dermal fibroblasts where TRPV4 is endogenously expressed [13].

The EC_50_ values of endocytosis may differ from those of the calcium influx. For example, previous studies reported EC_50_ values for internalization in HEK293 cells as 500 nM capsaicin for rat TRPV1 and as 31 nM GSK1016790A for human TRPV4 [17,18]. However, the present observations are in close agreement with the previous reports in terms of time dependency. Previous biochemical analyses demonstrated that rat TRPV1 expressed in HEK293 cells was internalized from the plasma membrane after 1 µM capsaicin stimulation for 10 min [17]. In contrast, mouse TRPV4 expressed in HeLa cells was internalized within 3 min after 10 nM GSK1016790A stimulation [19]. That is, similar internalization kinetics were observed with 3-fold higher concentrations of capsaicin and with 30-fold lower concentrations of GSK1016790A than used in the present study. The difference in the rate of endocytosis between TRPV1 and TRPV4 may be due to differences in intrinsic biological properties of TRPV channels rather than differences in the agonist potency. The different rates of endocytosis would be related to the physiological role of TRPV channels. Nociception induced by TRPV1 is essential for survival as a warning of impending damage to the organism [20]. Therefore, it would be important to keep the alarm on while the nociceptive stimulus is present, in contrast with the non-noxious sensory receptions.

The membrane domain localization of TRPV channels prior to agonist stimulation could be one of the properties that determine the rate of endocytosis. Both TRPV1 and TRPV4 have been reported to interact with caveolin-1, a marker of caveolae [21,22]. A clathrin-dependent endocytic pathway has also been reported for TRPV1 and TRPV4 [23,24]. Therefore, the reductions in TRPV channel density observed in the present study are probably attributable to caveolae- and clathrin-mediated endocytosis. The MSD–Δt plots of the fast and medium states suggest a simple diffusion mode, and it is unlikely to be directly involved in caveolae- or clathrin-mediated endocytosis. In contrast, the MSD–Δt plots of the immobile state suggest that the TRPV channels are localized in a membrane domain with a confinement length of ~150 nm, which corresponds to the average sizes of caveolae (60–80 nm diameter) and clathrin-coated pits (150–200 nm diameter) [25,26]. Therefore, endocytosis may occur after the entrapment of the molecules in the immobile domain. Notably, the proportions of higher-order oligomers decreased in the medium mobile state, which is accompanied by endocytosis (Figure 4G,H). It is expected that the higher-order oligomers of the TRPV channels are selectively recruited into the immobile domain, followed by their endocytosis. TRPV4 was pre-accumulated into the immobile domains compared to TRPV1, which would enable rapid endocytosis upon agonist stimulation. Since the membrane domain localization of TRPV channels before stimulation is independent of the activation process, one can speculate that it similarly regulates the rate of endocytosis after activation by other stimuli such as thermal or mechanical stimulation. This hypothesis needs to be tested in future studies.

The increase in the immobile state fraction of the TRPV1 channels upon activation (Figure 4A) is consistent with the previous TIRFM measurements, which compared the normalized MSDs before and immediately after capsaicin stimulation. Senning et al. [7] showed moving sparklets of a fluorescent Ca^2+^ indicator of TRPV1, suggesting that the active TRPV1 channels diffused laterally in the plasma membrane. In contrast, the sparklets indicated that the TRPV4 channels were predominantly immobile [27]. These differences can be explained partly by the diffusion state ratios before agonist stimulation observed in the present study (Figure 4E,F). Because the percentage of the fast state of TRPV1 was about twice that of TRPV4, TRPV1 is more likely than TRPV4 to open its gate in the fast state before it is trapped into the immobile domain.

The entrapment of TRPV1 in the immobile domain may contribute not only to the compartmentalized Ca^2+^ signaling but also to the sensitization of the channel. Previous studies suggested that protein kinases C [28], protein kinase A [29], Ca^2+^/calmodulin-dependent kinase II [30], and phosphatidylinositol 4,5-biphosphate (PIP_2_) [31] tend to localize to cholesterol/sphingomyelin rich membrane domains (lipid raft), including caveolae. It can be assumed that the recruitment of TRPV1 into the immobile domain immediately after agonist stimulation promotes the phosphorylation of TRPV1 by these kinases and its interaction with PIP_2_, which sensitizes the channel activity [32,33,34,35]. In consistent with this hypothesis, the cholesterol depletion from the cell membrane suppressed the activity of TRPV1 [36].

The previous electrophysiological study demonstrated that the acute Ca^2+^-mediated desensitization of rat TRPV1 in HEK293 cells [34]. The rate of acute desensitization by 1 μM capsaicin was much faster than endocytosis, and the desensitized TRPV1 was fully reactivated by 30 μM capsaicin stimulation. Therefore, the desensitization would be derived from a reduced apparent affinity for agonists, as discussed in the previous study [34], rather than endocytosis. The calcineurin-dependent dephosphorylation of TRPV1 [37] and the PIP_2_ depletion from the membrane [35] was proposed as mechanisms of the desensitization. Considering that the calcineurin was predominantly present in non-lipid raft domains [38], the present results propose a hypothesis that the membrane domain localization of TRPV1 spatially controls the balance between the Ca^2+^-dependent phosphorylation and dephosphorylation. The VB-HMM analysis suggested that TRPV1 molecules are in dynamic transition among three diffusion states with subsecond time constants (Appendix A). Assuming that the ratios of the three diffusion states of TRPV1 are restored after removing capsaicin, we would expect a decrease in the probability of TRPV1 interacting with the kinases and PIP_2_ in the lipid raft domain and an increase in dephosphorylation by calcineurin in the non-lipid raft domain. How the change in membrane composition affects the channel activity and the diffusion dynamics is currently unclear and will be an important issue in future studies.

A previous N&B analysis suggested that the dimers of TRPV1 form tetramers in the plasma membrane upon activation [6]. The increase in the proportion of immobile particles, most of which were larger than apparent tetramers, is qualitatively consistent with the previous report. However, it is unlikely that most of TRPV1 are in a dimeric form before capsaicin stimulation. The intensity histograms for TRPV1 showed a broad distribution, including molecules larger than tetramers in each diffusion state, even with <50% staining of the HaloTag (Figure 3H). Furthermore, ~20% of the particles stained with the different fluorescent dyes colocalized with each other (Figure 5E,F and Appendix A). Therefore, we infer that a significant proportion of TRPV1 molecules exists as tetramers or higher-order oligomers, even before capsaicin stimulation. TRPV4 channels showed broader intensity histograms, suggesting that TRPV4 is more likely to form higher-order oligomers than TRPV1 (Figure 3I). Generally, molecules with more transmembrane helices are thought to diffuse more slowly in the plasma membrane [39]. Therefore, this difference in oligomerization is probably related to the difference in the diffusion state ratios of TRPV1 and TRPV4.

In conclusion, the present comparative analysis of TRPV1 and TRPV4 channels demonstrates the different dynamics of their diffusion and oligomeric state changes upon activation. Before stimulation, TRPV4 is more likely to form higher-order oligomers in an immobile membrane domain than is TRPV1. The pre-accumulation of TRPV4 in the immobile domain is related to its rapid endocytosis. The selective loss of higher-order oligomers from the membrane increases the proportion of the fast mobile state of TRPV4. Similar changes also occur in TRPV1 channels upon activation, but at a slower rate because its recruitment to the immobile domain is required before endocytosis.

## 4. Materials and Methods

### 4.1. Materials

Capsaicin and GSK1016790A were purchased from Cayman Chemical (Ann Arbor, MI, USA).

### 4.2. Construction of cDNA

The human TRPV1 (a gift kindly supplied by Asuka Inoue, Tohoku University) and human TRPV4 coding sequences (cloned from the MegaMan Human Transcriptome Library) were amplified with PCR and inserted into the pFN21A HaloTag^®^ CMV Flexi^®^ Vector (Promega, Madison, WI, USA) to fuse the HaloTag to the N-terminus of the expressed protein. The CD86 (M1-R277) coding sequence was amplified with PCR and inserted into the pEGFP-N1 mammalian expression vector (Clontech, CA, USA), in which the coding sequence of enhanced green fluorescent protein (EGFP) was substituted with that of HaloTag7.

### 4.3. Cell Culture and Generation of TRPV4-KO HEK293 Cell Line with the CRISPR/Cas9 System

HEK293A cells (a gift kindly supplied by Asuka Inoue, Tohoku University) were cultured in Dulbecco’s modified Eagle’s medium (DMEM) containing phenol red (Wako Chemicals, Tokyo, Japan) supplemented with 10% fetal bovine serum at 37 °C under 5% CO_2_. The single guide RNA (sgRNA) sequence targeting the region adjacent to exon 14 of TRPV4 (GAGCAGCACCAAGTACCCCG) was designed using the online CRISPR design tool from the Zhang laboratory at the Broad Institute (Cambridge, MA, USA). The sgRNA sequence was then cloned into the pSpCas9−2A-Puro (PX459) vector (a gift from Feng Zhang, Addgene plasmid #62988). HEK293 cells were transfected with the vector using Lipofectamine 3000 (Invitrogen, Waltham, MA, USA). At 48 h after transfection, the cells were selected with fresh medium containing 1 µg/mL puromycin, and were then maintained in DMEM for 7 days to allow colony formation. Six individual colonies were picked and seeded manually into six-well plates in DMEM. The colonies were allowed to expand for 10–14 days, and the clones in the six-well plates were then duplicated for either cell culture or genomic DNA extraction. Genomic DNA was extracted with the GenElute™ Mammalian Genomic DNA Miniprep Kit (Sigma-Aldrich, St. Louis, MO, USA), according to the manufacturer’s manual. The CRISPR/Cas9-targeted sequence was PCR amplified (forward primer: 5′-GCAATGAGGACCAGACCAACTGCA-3′ and reverse primer: 5′-CCAGATGTGCTTGCTCTCCTTG-3′), and the −1 deletion was confirmed with DNA sequencing (Figure 1C).

### 4.4. Single-Molecule Imaging

TRPV4-KO HEK293 cells were transfected with plasmid DNA expressing HaloTag-fused TRPV1 or TRPV4 on glass coverslips (Matsunami, Osaka, Japan) in a 60 mm dish 1 day before imaging. Lipofectamine 3000 (Invitrogen) was used for transfection. After incubation for 15 min at room temperature, the transfection mixture containing plasmid DNA (0.15 µg), P3000 reagent (0.30 µL), Lipofectamine 3000 reagent (2.5 µL), and Opti-MEM (60 µL, Gibco) was added to cells cultured with DMEM (3 mL) in a 60 mm dish. After overnight incubation, the HaloTag-fused TRPV1 or TRPV4 was labeled with 30 nM HaloTag SF650 ligand (Goryo Chemical, Inc., Hokkaido, Japan) or 30 nM HaloTag JF549 (Promega) in DMEM without phenol red for 15 min at 37 °C under 5% CO_2_. The HaloTag-ligand-treated HEK293 cells on coverslips were washed three times with DMEM without phenol red (3 mL) in a 60 mm dish. The coverslips were mounted in a metal chamber (Invitrogen) and washed five times with 400 µL of Hanks’ balanced salt solution (HBSS; Sigma-Aldrich) containing 15 mM HEPES (pH 7.3) and 0.01% bovine serum albumin (BSA) without NaHCO_3_. Ligand (100 µL of compound solution at 5 times the final concentration) or vehicle solution (100 µL) was added to the chamber with 0.01% BSA/HBSS (400 µL), and movies were acquired twice at a steady state. Single-molecule imaging was performed over 25 min at room temperature (25 °C).

The fluorescently labeled TRPV1 or TRPV4 on the basal cell membrane was observed with total internal reflection illumination with an inverted fluorescence microscope (TiE, Nikon, Tokyo, Japan). The cells were illuminated with a 532 nm, 100 mW laser (Compass 315M-100) with an ND50 filter for JF549 and with a 637 nm, 140 mW laser (OBIS 637, Coherent) for SF650, through an objective (PlanApo 60×, NA 1.49, Nikon) with a dichroic mirror (ZT532/640rpc, Chroma) for JF549 and SF650. The light emitted from JF549 or SF650 was split into two light paths by a two-channel imaging system (M202J, Nikon) with a dichroic mirror (FF640-FDi01, Semrock for SF650) and was simultaneously detected with two electron-multiplying (EM)-CCD cameras (ImagEM, Hamamatsu) after passing through band-pass filters (FF01-585/40 (Semrock) for JF549 and FF01-676/29 (Semrock) for SF650). A 4× relay lens was placed before the two-channel imaging system to magnify the image (67 nm/pixel). The fluorescent images were recorded with AutoImagingSystem (AIS, Zido) with the following settings: exposure time, 30.5 ms for dual-color imaging; electron-multiplying gain, 200; spot noise reduction, on. The multiple TIFF files (16 bit) were processed using ImageJ analysis software (National Institutes of Health, Bethesda, MD, USA), as follows. The background was subtracted with a rolling ball radius of 25 pixels, followed by two-frame averaging of the images with the Running_ZProjector plugin (available at Vale Lab home page; https://valelab4.ucsf.edu/~nstuurman/IJplugins/, accessed on 1 April 2021). Single-molecule tracking and the VB-HMM analysis were conducted with AutoAnalysisSystem (AAS, Zido). The parameters of the trajectories were calculated, the curves were fitted, and the illustrations in the figures were drawn with smDynamicsAnalyzer, an open-source macro of Igor Pro 8.0 (WaveMetrics), as previously described in detail [9] (available at https://github.com/masataka-yanagawa/IgorPro8-smDynamicsAnalyzer, accessed on 1 April 2021.). The colocalization of the JF549-labeled molecules and SF650-labeled molecules was defined as a distance of <200 nm between different molecules in the same frame and diffusion state, as estimated with the VB-HMM analysis. The on-rate constant was calculated as $\frac{V_{0}}{\left[Ch1\right]\left[Ch2\right]}$, where V_0_ is the initial rate of the cumulative number of association events per unit area, [Ch1] and [Ch2] are the particle densities for each channel.

### 4.5. FLIPR Calcium Mobilization Assay

The pharmacological activation of wild-type or HaloTag-fused TRPV1 and TRPV4 was analyzed with the FLIPR^®^ Calcium 6 Assay Kit (Molecular Devices, #R8191 bulk kit, San Jose, CA, USA), according to the manufacturer’s instructions. TRPV4-KO HEK293 cells were transfected with the plasmid DNA on assay plates (96-well black polymer BTM collagen plates; Thermo Fisher Scientific, Waltham, MA, USA). The transfection mixture (0.15 µg of plasmid DNA, 0.30 µL of P3000 reagent, 5.0 µL of Lipofectamine 3000, and 120 µL of Opti-MEM) was added to cells cultured in DMEM (12 mL) on assay plates. After overnight incubation, the cells were incubated for 1.5 h with the calcium indicator (FLIPR^®^ Calcium 6 Assay Kit) in 80 µL of HBSS at 37 °C under 5% CO_2_ and then for 30 min at room temperature. Calcium influx was detected with a microplate reader (FlexStation3, Molecular Devices) with the following parameters: mode, fluorescence; excitation 485 nm, shutdown 515 nm, emission 525 nm; photomultiplier gain, automatic; flashes per reading, 6; read from the bottom. Ligand (20 µL of compound solution at 5 times the final concentration) was added 30 s after the start of measurements. The calcium influx trace and the dose-response curves were analyzed using the peak fluorescence intensity normalized to the fluorescence intensity before stimulation. The dose-response curves were fitted with the Hill Equation (1) with Igor Pro 8.0 (WaveMetrics).
(1)$$f\left(x\right)=bottom+\frac{top-bottom}{1+{\left(\frac{EC_{50}}{x}\right)}^{n}}$$
where *n* is the Hill coefficient.

### 4.6. Saturation Binding Assays of JF549 and SF650

TRPV4-KO HEK293 cells were transfected with the plasmid DNA encoding wild-type or HaloTag-fused TRPV1 or TRPV4 on assay plates (96-well black polymer BTM collagen plates; Thermo Fisher Scientific). The transfection mixture (1.0 µg of plasmid DNA, 2.0 µL P3000 reagent, 2.5 µL of Lipofectamine 3000, and 60 µL of Opti-MEM (Gibco)) was added to cells cultured with DMEM (6 mL) on assay plates. After overnight incubation, wild-type or HaloTag-fused TRPV1 or TRPV4 was labeled with 0.3–300 nM HaloTag JF549 (Promega) or 0.3–300 nM HaloTag SF650 ligand (Goryo Chemical, Inc.) in DMEM without phenol red for 15 min at 37 °C under 5% CO_2_. The HaloTag-ligand-treated cells on assay plates were washed three times with DMEM without phenol red. After incubation for 30 min at room temperature, the saturation binding of the HaloTag ligand was detected with a microplate reader (FlexStation3, Molecular Devices) with the following parameters: mode, fluorescence; excitation 530 nm, cutoff 570 nm, emission 580 nm for JF549 and excitation 640 nm, cutoff 665 nm, emission 675 nm for SF650; photomultiplier gain, automatic; flashes per read, 6; read from bottom. The background fluorescence intensity was estimated from the intensity of cells untreated with the HaloTag ligand. Nonspecific binding was determined from the fluorescence intensity of wild-type TRPV1- or TRPV4-transfected cells. Specific binding was calculated as the difference between the total binding to cells expressing HaloTag-fused TRPV1 or TRPV4 and nonspecific binding. The data were fitted with the Hill Equation (1).

## Figures and Tables

**Figure 1 ijms-22-08473-f001:**
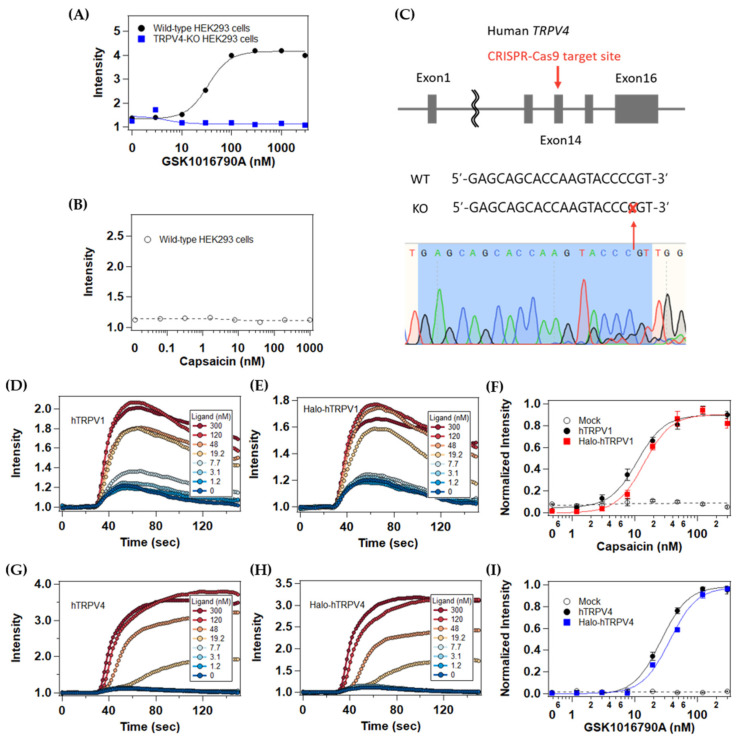
Evaluation of the effect of HaloTag fusion on human TRPV1 and TRPV4. (**A**) GSK1016790A dose-response curve in mock-transfected wild-type HEK293 cells (black) and mock-transfected TRPV4-KO HEK293 cells (blue). (**B**) Capsaicin dose-response curve in mock-transfected wild-type HEK293 cells (circle). (**C**) Sequencing results for the sgRNA target site in the human TRPV4 gene. (**D**,**E**) Representative capsaicin-induced Ca^2+^ influx in TRPV4-KO HEK293 cells transfected with plasmid expressing wild-type hTRPV1 (**D**) or HaloTag-fused hTRPV1 (**E**). (**F**) Capsaicin dose-response curve in mock-transfected cells (circle), wild-type-hTRPV1-expressing cells (black filled circle), or Halo-Tag-fused-hTRPV1expressing cells (red). (**G**,**H**) Representative GSK1016790A-induced Ca^2+^ influx in TRPV4-KO HEK293 cells transfected with plasmid expressing wild-type hTRPV4 (**G**) or HaloTag-fused hTRPV4 (**H**). (**I**) GSK1016790A dose-response curves of mock-transfected cells (circle), wild-type-hTRPV4-expressing cells (black filled circle), and HaloTag-fused-hTRPV4-expressing cells (blue). Data in (**F**,**I**) are means ± SEM of three independent experiments.

**Figure 2 ijms-22-08473-f002:**
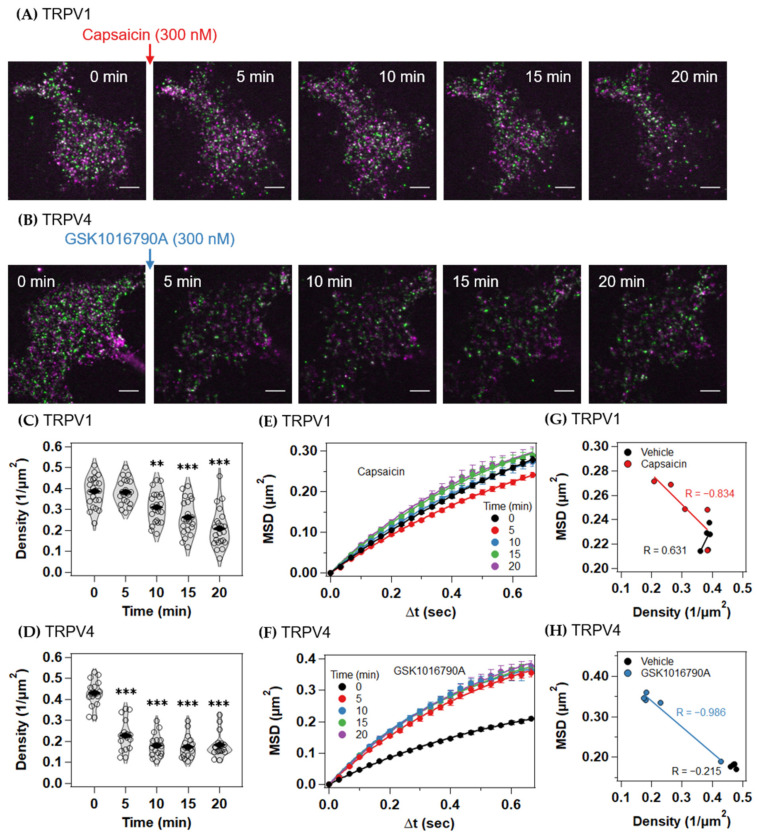
TIRFM image, receptor density, mean squared displacement (MSD)–Δt plots of TRPV1 and TRPV4 channels after activation. (**A**,**B**) Representative time-lapse TIRFM images of TRPV4-knockout (KO) HEK293 cell expressing JF549 (magenta)- and SF650 (green)-labeled TRPV1 after capsaicin (300 nM) stimulation (**A**); and TRPV4 after GSK1016790A (300 nM) stimulation (**B**). Scale bars, 5 µm. (**C**,**D**) Density of JF549-labeled TRPV1 (**C**) and JF549-labeled TRPV4 (**D**) after agonist stimulation. (**E**,**F**) MSD–Δt plots of the trajectories under the indicated ligand conditions. JF549-labeled TRPV1 (**E**), JF549-labeled TRPV4 (**F**). Data are means ± SEM of 17–22 cells. ** *p* < 0.01, *** *p* < 0.001 (one-way ANOVA followed by Dunnett’s multiple-comparisons test versus basal level). (**G**,**H**) Correlation between receptor density and time-averaged MSD at each time point for JF549-labeled TRPV1 (**G**) and JF549-labeled TRPV4 (**H**). Each circle represents a time point. Lines are regression lines, and R is Pearson’s correlation coefficient.

**Figure 3 ijms-22-08473-f003:**
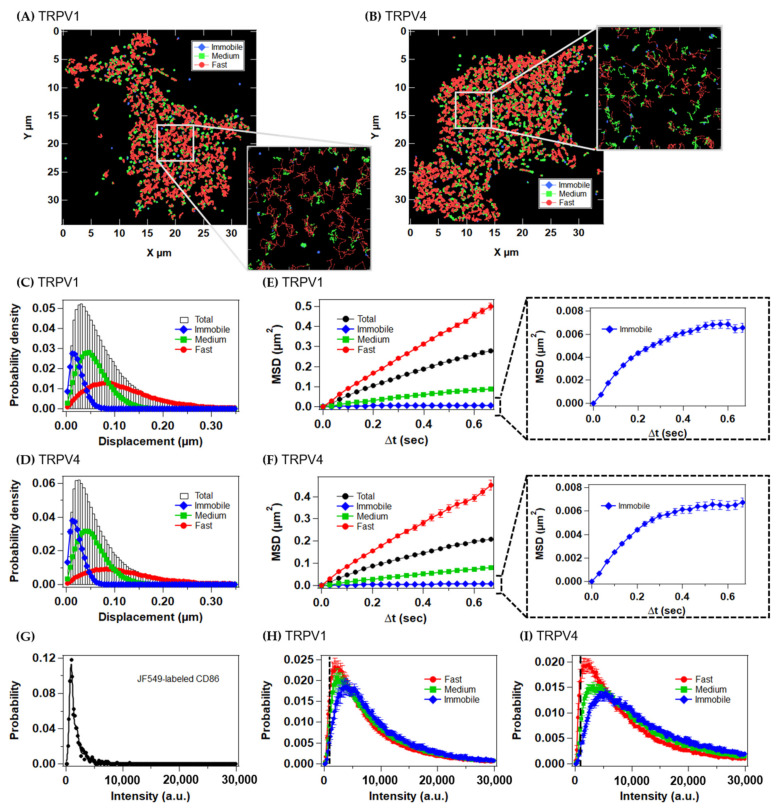
VB-HMM analysis of the trajectories of TRPV1 and TRPV4 molecules. (**A**,**B**) Each step in the trajectories was categorized into three diffusion states. The immobile, medium, and fast states are shown in blue, green, and red, respectively. (**C**,**D**) Histogram of the displacement of JF549-labeled TRPV1 (**C**) and TRPV4 (**D**) at the basal level. (**E**,**F**) MSD–Δt plots of each diffusion state of JF549-labeled TRPV1 (**E**), JF549-labeled TRPV4 (**F**). Data are means ± SEM of 17–22 cells. (**G**) Intensity histograms of JF549-labeled CD86 measured under the same conditions as described for TRPV1 and TRPV4. (**H**,**I**) Histogram of the intensity of JF549-labeled TRPV1 (**H**) and JF549-labeled TRPV4 (**I**) at the basal level. Dashed line represents the intensity of single molecules (896 a.u.) estimated from JF549-labeled CD86 (the monomeric control).

**Figure 4 ijms-22-08473-f004:**
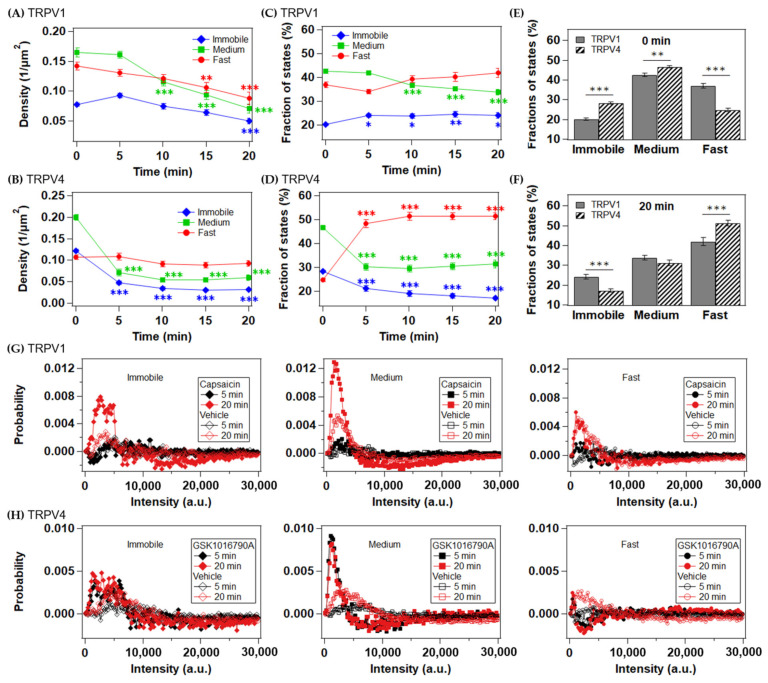
Agonist-induced changes in the membrane dynamics of the TRPV1 and TRPV4 channels. (**A**,**B**) Time-dependent changes in the density of each diffusion state of JF549-labeled TRPV1 upon capsaicin (300 nM) stimulation (**A**) and TRPV4 upon GSK1016790A stimulation (**B**). (**C**,**D**) Time-dependent changes in the fractions of the diffusion states of JF549-labeled TRPV1 (**C**) and JF549-labeled TRPV4 (**D**) after activation. The immobile, medium, and fast states are shown in blue, green, and red, respectively. Data are means ± SEM of 17–22 cells. ** *p* < 0.01, *** *p* < 0.001 (one-way ANOVA followed by Dunnett’s multiple-comparisons test versus basal level). (**E**,**F**) Comparison of the diffusion state fractions of TRPV1 and TRPV4 at 0 min (**E**) and 20 min (**F**). ** *p* < 0.01, *** *p* < 0.001; two-tailed *t*-test when compared with the same diffusion state of TRPV1. (**G**,**H**) Time-dependent changes in the intensity of JF549-labeled TRPV1 (**G**) and JF549-labeled TRPV4 (**H**) after stimulation with vehicle (black) or agonist (red).

**Figure 5 ijms-22-08473-f005:**
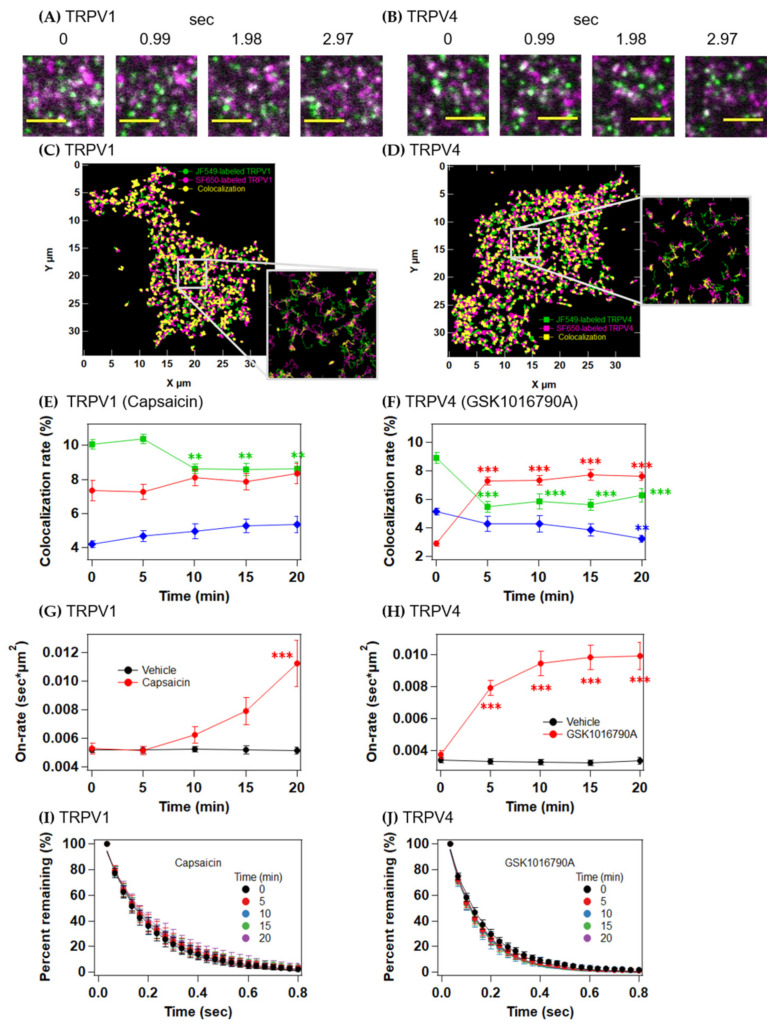
Colocalization analysis of TRPV1 and TRPV4 upon activation. (**A**,**B**) Representative images of the colocalization of JF549-labeled TRPV1 (green) and SF650-labeled TRPV1 (magenta) (**A**) and JF549-labeled TRPV4 (green) and SF650-labeled TRPV4 (magenta) (**B**). Colocalized bright spots appear to be white. Scale bars, 3 µm. (**C**,**D**) Trajectories of colocalized TRPV1 molecules (**C**) and TRPV4 molecules (**D**). (**E**,**F**) Time-dependent changes in the colocalized fractions of the diffusion states. TRPV1 after capsaicin stimulation (**E**), TRPV4 after GSK1016790A stimulation (**F**). (**G**,**H**) On-rate constants of JF549-labeled TRPV1 (**G**) and JF549-labeled TRPV4 (**H**) after stimulation with vehicle (black) or agonist (red). (**I**,**J**) Time-dependent changes in the percentage of TRPV1 remaining (**I**) and TRPV4 remaining (**J**) upon activation. Data are means ± SEM of 17–22 cells. ** *p* < 0.01, *** *p* < 0.001 (one-way ANOVA followed by Dunnett’s multiple-comparisons test versus the basal level (0 min)).

**Figure 6 ijms-22-08473-f006:**
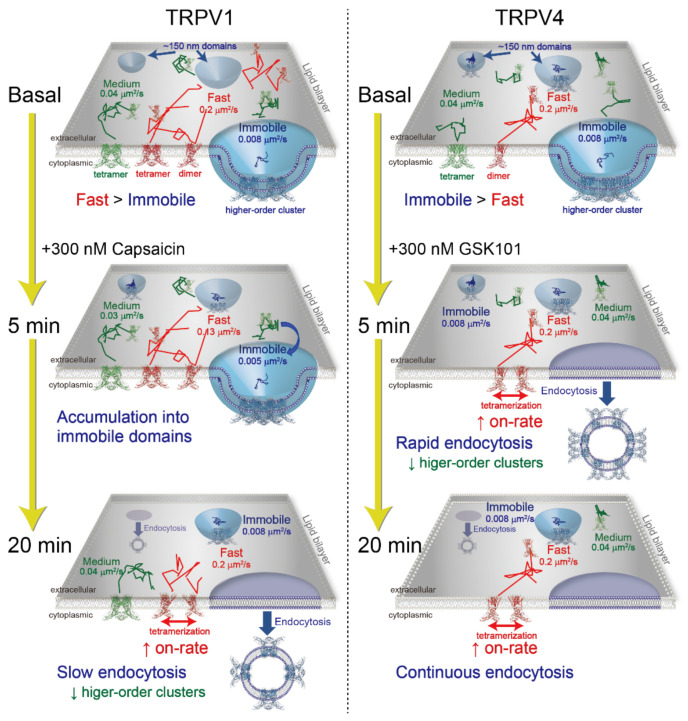
Schematic representation of TRPV1 and TRPV4 channel diffusion in the plasma membrane after activation. The equilibrium among the three diffusion states (immobile, medium, and fast) and the particle densities of TRPV1 and TRPV4 changed temporarily upon agonist stimulation.

## Data Availability

All data is contained within the article.

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
