# Peer review of "Comparative Analysis of Single-Molecule Dynamics of TRPV1 and TRPV4 Channels in Living Cells"

_ijms, 2021, doi:10.3390/ijms22168473_

Round 1

Reviewer 1 Report

In this article, Kuwashima et al. report their results on the time-dependent changes in the single-molecule dynamics of two members of the vanilloid transient receptor potential family, TRPV1 and TRPV4. The authors present evidence that these two proteins exhibit differences in the dynamics before and after stimulation with their respective agonists.

Overall, the study contains original and interesting data and the experiments are carefully and correctly performed. It is also valuable that, unlike in previous studies, the two channels are compared using the same technical approach. The results are not only of potential interest to the TRP channel community (It's amazing to learn something we don't already know about such intensively studied channels), but are also important from a methodological perspective.

I have only two interrelated comments that I would like to ask the authors to comment on in the discussion:

  • The authors used a saturating concentration (300 nM) for both agonists. How could the equilibrium among the three diffusion states depend on the agonist concentration? Although the authors demonstrate that both ligands have approximately similar efficacy on TRPV1 and TRPV4, the reported half-effective concentration pEC50 is 7.5 for capsaicin and 8.7 for GSK1016790A (i.e. a difference of one order).
  • For TRPV1, it has been proposed that its endocytosis may underlie functional desensitization of the channels upon longer exposure to capsaicin. It has been previously demonstrated by Novakova-Tousova et al. (Neuroscience 149, 2007, 144–154) that desensitized channels can be fully reactivated by utilizing a supersaturating concentration of agonist (capsaicin 30 microM or piperine 100 microM). Could the authors find a hypothesis to explain this observation?

Minor:

  • Page 8, Line 247: „TRPV4“ instead of „TPRV4“
  • Page 10, Line 360 and Page 11, Line 404: What does it mean 5x concentration of a ligand?
  • Page 11, the Hill equation (1) is incorrect (in the denominator, it should be (1+(EC50/x)^n)

Reviewer 2 Report

Kuwashima et al. have examined the dynamics of TRPV1 and TRPV4 channels and provide interesting new insights as regards their spatial and temporal distribution in response to pharmacological agonists.

Comments:

  1. As regards the role of TRPV1 and TRPV4 it might be better, from the physiological perspective, to say that TRPV1 is activated by noxious thermal stimuli, whereas TRPV4 is activated by non-noxious thermal stimuli (rather than mild).
  2. Differences in the rate of endocytosis of TRPV1 and TRPV4 are very interesting. However, could authors comment whether this difference arises from differences in intrinsic biological properties of TRPV1 and TRPV4 or is this difference specific for the ligand used to activate these receptors? (Would the difference be similar with another stimulus?)
  3. In addition, would authors expect a different temporal (faster) response if higher capsaicin concentration is used? Both agonists were used in equimolar concentrations, but do they have similar potency on their receptors? It would be interesting to know, whether time-dependent responses change with different concentrations of TRPV1 and TRPV4 agonists.
  4. It would be nice if authors could discuss briefly whether different rates of endocytosis would be expected also with thermal stimuli or is different dynamics only a response to chemical stimulation.
  5. It would be interesting to speculate (in the Discussion) what are the functional consequences of different rates of endocytosis of TRPV1 and TRPV4. For instance, could slower rate of endocytosis of TRPV1 be related to a slower adaptation of nociceptors compared with some other receptors.
  6. The authors have produced a lot of interesting supplementary data. It would be useful to move some of these results in the main manuscript. (To avoid reader having to refer to supplement so much.)
